# Disposition Decision Support by Laboratory Based Outcome Prediction

**DOI:** 10.3390/jcm10050939

**Published:** 2021-03-01

**Authors:** Oliver S. Mueller, Katharina M. Rentsch, Christian H. Nickel, Roland Bingisser

**Affiliations:** 1Emergency Department, University Hospital Basel, 4031 Basel, Switzerland; oliversimon.mueller@usb.ch (O.S.M.); christian.nickel@usb.ch (C.H.N.); 2Laboratory Medicine, University Hospital Basel, 4031 Basel, Switzerland; katharina.rentsch@usb.ch

**Keywords:** decision support, laboratory, outcome prediction, emergency medicine, triage, disposition, mortality, emergency severity index

## Abstract

Disposition is one of the main tasks in the emergency department. However, there is a lack of objective and reliable disposition criteria, and diagnosis-based risk prediction is not feasible at early time points. The aim was to derive a risk score (TRIAL) based on routinely collected baseline (TRIage level and Age) and Laboratory data—supporting disposition decisions by risk stratification based on mortality. We prospectively included consecutive patients presenting to the emergency department over 18 weeks. Data sets of routinely collected baseline (triage level and age) and laboratory data were used for multivariable logistic regression to develop the TRIAL risk score predicting mortality. Routine laboratory variables and disposition cut-offs were chosen beforehand by expert consensus. Risk stratification was based on low risk (<1%), intermediate risk (1–10%), and high risk (>10%) of in-hospital mortality. In total, 8687 data sets were analyzed. Variables identified to develop the TRIAL risk score were triage level (Emergency Severity Index), age, lactate dehydrogenase, creatinine, albumin, bilirubin, and leukocyte count. The area under the ROC curve for in-hospital mortality was 0.93. Stratification according to the TRIAL score showed that 67.5% of all patients were in the low-risk category. Mortality was 0.1% in low-risk, 3.5% in intermediate-risk, and 26.2% in high-risk patients. The TRIAL risk score based on routinely available baseline and laboratory data provides prognostic information for disposition decisions. TRIAL could be used to minimize admission in low-risk and to maximize observation in high-risk patients.

## 1. Introduction

The three major tasks of emergency medicine are triage, work-up, and disposition [1].

If disposition (e.g., discharge, admission, or intensive care) is not given by logistic factors, such as the need for intervention, it is based on outcome prediction, as preventing admission of low-risk patients has become similarly important to preventing discharge of patients at risk of early deterioration or death.

As the economic pressure is rising and unexpected death after discharge is infrequent [2], unnecessary additional cost of hospitalization has moved to the center of attention in many health care systems. Resources used for hospitalized patients may be up to tenfold higher as compared to ambulatory care in comparable situations [1]. In spite of the existing research on outcome prediction (based on routine administrative data [3], vital signs [4,5,6,7], observation [8], machine learning [9], combinations of age, vital signs, and loss of independence [10], or admission laboratory data without [11] or with inclusion of patient age [12,13,14] or vital signs [15]), only a few studies have focused on the direct support of disposition decisions [8,16,17,18]. As all of these attempts have drawbacks and disease-specific tools rely on a sound diagnosis, a call for the development of a “Universal Safe to Discharge Score” was recently issued [19].

We have therefore aimed at improving disposition decisions using laboratory-based outcome prediction in combination with age and level of acuity, as these two factors contain the best baseline prognostic information [20,21,22]. Our primary objective was to develop a risk score based on the predictive performance of baseline and laboratory data in order to stratify patients into groups of low, intermediate, and high risk of short-term mortality.

## 2. Materials and Methods

### 2.1. Study Design and Setting

This is a secondary analysis of prospectively collected data of the EMERGE study, which serves as a quality-control study. The study protocol was approved by the local ethics board (EKNZ-236/13; www.eknz.ch, accessed date: 12 December 2019) and was conducted at the emergency department (ED) of the University Hospital Basel, Switzerland with a census of over 50,000 yearly presentations.

All patients presenting to the ED of the University Hospital Basel were eligible. Data were acquired consecutively from 21 October to 11 November 2013, 1 February to 23 February 2015, 30 January to 19 February 2017, and 18 March to 20 May 2019. By selecting different observation periods, we wanted to consider potential seasonal effects.

### 2.2. Selection of Participants

All consecutive patients presenting to the ED were screened. Patients who denied consent were not included. Pediatric, obstetric, and ophthalmology patients were treated at nearby facilities.

### 2.3. Data Collection

All patients presenting to the emergency department were registered by a member of the study team and an electronic health record (EHR) was opened. The study team consisted of trained medical students. Patients were screened 24 h a day, 7 days a week.

Information about demographics (age, gender), Emergency Severity Index (ESI) triage level, disposition (discharge, hospitalization, and intensive care unit admission), in-hospital mortality, and laboratory data were extracted from the EHR.

Only patients with full data sets were analyzed. We excluded patients with missing triage or laboratory data.

#### 2.3.1. Triage

A triage nurse or emergency physician triaged all patients according to the German version of the ESI (version 4) [20].

ESI categorizes patients into five levels. High acuity patients are assigned to ESI level 1 (most urgent) or ESI level 2, while less acute patients are classified according to the expected number of resources required (none in ESI level 5 and more than one in ESI level 3) [23].

#### 2.3.2. Laboratory Data

Laboratory data of the first examinations after presentation were used for all analyses. No additional tests were performed for study purposes only. If multiple blood samples were taken from the same patient, only results from the first set were analyzed. Cases were excluded if blood samples arrived at the laboratory later than 24 h after presentation, or if analyses were incomplete (for analysis: added to patient group “without laboratory data”).

To ensure usefulness in everyday clinical practice, we restricted our investigation to routinely available laboratory parameters. The laboratory variables to be examined were determined beforehand by expert consensus. The decision was based on clinical relevance and previous publications [11,12,13,14,24]. Selected variables were sodium (mmol/L), potassium (mmol/L), urea (mmol/L), creatinine (μmol/L), bilirubin (μmol/L), aspartate transaminase (U/L), lactate dehydrogenase (U/L), albumin (g/L), C-reactive protein (mg/L), glucose (mmol/L), leukocytes (×10^9^/L), thrombocytes (×10^9^/L), and hemoglobin (g/L).

Please refer to the Appendix A for details on laboratory assessments.

### 2.4. Outcomes

The primary outcome was the validity of the risk score predicting the probability of in-hospital mortality (death between presentation to the ED and discharge). Secondary outcomes were hospitalization and intensive care unit (ICU) admission.

Hospitalization was defined as admission to any hospital ward from the ED with a minimum of one overnight stay.

ICU-admission was defined as transfer to a medical or surgical ICU, intermediate care unit, or stroke unit during index hospitalization.

### 2.5. Data Analysis

Descriptive analyses are presented as medians with interquartile ranges for continuous variables and counts with percentages for categorical variables. Baseline characteristics and outcomes of patients included were compared to those of patients without laboratory data and *t*-test and chi-squared test were used to assess differences. Furthermore, the distribution of outcomes across the ESI levels was investigated. We used the statistical software R (Version 3.6.1, R Foundation for Statistical Computing, FreeSoftware Foundation, Boston, MA, USA) for all predictive calculations.

#### 2.5.1. Derivation of Predictive Models

Four models were developed using multivariable logistic regression and their predictive performances for probability of in-hospital mortality were compared. (A) The baseline model included demographic data (age and gender) and triage category (ESI). (B) The laboratory model included all laboratory parameters tested (sodium, potassium, urea, creatinine, bilirubin, aspartate transaminase, lactate dehydrogenase, albumin, C-reactive protein, glucose, leukocytes, thrombocytes, and hemoglobin). All laboratory parameters except sodium and hemoglobin were log-transformed due to asymmetrical distribution. Restricted cubic splines [25] were used for all laboratory parameters to account for nonlinear relationships between the individual variables and in-hospital mortality on the log-odds scale. These variables with nonlinear relationships to the outcome were inserted in a logistic regression model and based on the log-odds the probability of in-hospital mortality was calculated. (C) The full model included all laboratory and baseline parameters, combining models A and B. (D) The reduced model included a reduced set of laboratory and baseline data, reducing the complexity of model C. Based on the full model, the combination of laboratory and baseline parameters with the best predictive validity was selected using stepwise backwards elimination [26]. We used AIC (Akaike information criterion) as a selection criterion. Backwards elimination starts with the full model and at each step the variable which increases the AIC the most gets eliminated. It stops if any of the remaining variables would decrease the AIC. The aim was to create a score with a minimal number of variables without losing predictive power.

Discriminatory validity of the models was evaluated with receiver operating characteristic (ROC) curves and their areas under the curve (AUC) with 95% confidence intervals (CI). DeLong’s test [27] was used to compare the models. The AUCs were adjusted for overfitting using bootstrapping. Additionally, we performed a 10-fold internal cross validation.

Calibration of the models was assessed by plotting the predicted probabilities versus the actual probabilities of in-hospital mortality.

#### 2.5.2. Development of the TRIAL Score

For clinical usability, we chose the acronym TRIAL containing the three main categories (TRIage level, Age, Lab). Using the selected variables of the reduced model, we created a nomogram (calculating outcome probabilities based on predictor variables). Based on nonlinear logistic regression, predictor variables were mapped on a point scale according to the relative contribution of their values to the predicted in-hospital mortality. The likelihood of in-hospital mortality was predicted by the number of points a patient received.

Two risk cut-offs were selected by experts based on clinical usefulness for disposition decisions. They were set at 1% and 10% probability of in-hospital mortality, stratifying to low risk (<1%), intermediate risk (1–10%) and high risk (>10%)—“low risk” representing patients safe for discharge, “intermediate risk” patients to be hospitalized, and “high risk” patients to be monitored in intermediate or intensive care units.

In addition, we evaluated the predictive performance of the variables of our score fitted for the secondary outcomes hospitalization and ICU-admission. Predictive validity was assessed with areas under the receiver operating characteristic curves.

## 3. Results

### 3.1. Characteristics

During the study period, 17,327 patients presented to the ED. Out of 14,440 enrolled patients, we excluded 17 patients due to missing ESI levels and 5736 patients without laboratory data. This resulted in 8687 patients (see Figure 1 for details).

Their median age was 61 years (IQR 42–78 years) and 51.5% were male. Most were classified as ESI 3 (51.7%) and ESI 2 (37.3%). In addition, 183 patients died in hospital (2.1%), 4623 patients were hospitalized (53.2%), and 758 patients (8.7%) received ICU care.

Patients without laboratory data were younger, had lower acuity ESI levels, and better outcomes (see Table 1).

Fourteen patients with missing or delayed laboratory analyses died (see Table A1).

Characteristics and outcomes were stratified by ESI level. In-hospital mortality gradually decreased from ESI 1 to 5. A similar decrease was observed for ICU-admissions and hospitalizations (see Table 2).

One ESI 4 patient died, and 10 ESI 4/5 patients were transferred to ICU (see Table A2).

### 3.2. Performance of Predictive Models

The predictive performance of the four models (baseline model, laboratory model, full model, and reduced model) regarding probability of in-hospital mortality is shown in Figure 2.

The baseline model’s AUC was 0.86 (95% CI 0.83–0.88), the laboratory model’s AUC was 0.90 (95% CI 0.88–0.92), and the full model’s AUC was 0.94 (95% CI 0.93–0.95). Stepwise backwards elimination reduced the full model to seven variables: Age, ESI, albumin, lactate dehydrogenase, leukocytes, creatinine, and bilirubin (nonlinear relationships to the probability of in-hospital mortality on the log-odds scale are shown in Figure A1). The reduced model’s AUC was 0.93 (95% CI 0.91–0.94).

After adjustment for overfitting AUCs was 0.85 for the baseline model, 0.88 for the laboratory model, 0.92 for the full model, and 0.91 for the reduced model. Results from 10-fold internal cross validation are shown in the Appendix A. Comparison of the models using DeLong’s test is shown in Table A3.

The reduced model predicted in-hospital mortality accurately up to a risk of 20% (see Figure A2). For higher probabilities of death, the reduced model showed an overestimation of the predicted in-hospital mortality. Of all patients, 132 (1.5%) had an estimated mortality of more than 20%.

### 3.3. TRIAL Risk Score

The TRIAL risk score is shown in Table 3. Results exactly between two values are to be assigned to the value contributing a higher number of points. The low-risk cut-off (1% mortality) is at 145 points, and the high-risk cut-off (10% mortality) at 176 points.

For the low-risk cut-off (1%), sensitivity was 0.96, specificity 0.71, negative predictive value 0.99, and positive predictive value 0.067 regarding mortality prediction. For the high-risk cut-off (10%), sensitivity was 0.54, specificity 0.97, negative predictive value 0.99, and positive predictive value 0.26 regarding mortality prediction.

Patients were stratified according to ESI (1–5) and risk of death (see Table 4).

Overall, 344 patients (4.0%) were assigned to high risk, 2478 (28.5%) to intermediate risk, and 5865 (67.5%) to low risk. Mortality was 26.2% in the high-risk group, 3.5% in the intermediate-risk group, and 0.1% in the low-risk group.

The distribution of patients across the risk groups stratified by ESI level was as follows: The majority of ESI 1 patients (54.1%) were assigned to the high-risk group (mortality 30.6%). High-risk was attributed to 111 patients with ESI 2 (3.4%) and 60 patients with ESI 3 (1.3%). Mortality was 20.7% (ESI 2) and 23.3% (ESI 3), respectively. No ESI 4 or 5 patients were classified as high-risk patients. The low-risk group comprised 67.5% of all patients and mortality was 0.1% (7 deaths of 5865 low-risk cases, see Table A4), which was similar across all ESI levels. Deaths in the laboratory-based low-risk group were found in aortic dissection (*n* = 3), COPD exacerbation (*n* = 1), pneumonia (*n* = 1), subdural hematoma (*n* = 1), and suicide (*n* = 1).

Predictive validity of the variables of the reduced model fitted for secondary outcomes was good with AUCs of 0.80 for hospitalization and 0.81 for intensive care (details in Table A5).

## 4. Discussion

Baseline and routine laboratory parameters may be used for outcome prediction supporting disposition decisions. The main findings of this study are that the predictive validity is excellent regarding mortality and good regarding intensive care, that two thirds of all patients can be attributed to the low-risk category, and that the derived TRIAL risk score could support early disposition decisions.

Health care systems around the globe have reported a public hospital bed crisis [28,29,30] and many approaches were considered, such as the implementation of guidelines to reduce hospital length of stay [31], comprehensive geriatric interventions [32], or early discharge [33], in order to reduce the economic burden of over-hospitalization. The absence of evidence-based guidelines for admission avoidance [33,34] may be due to a lack of published evidence focusing on disposition after emergency work-up. Because public hospitals tend to admit the majority of acutely ill patients from their emergency departments, they are directly affected by disposition decisions after emergency work-up.

The study’s results show that such disposition decisions may be facilitated using laboratory data because risk stratification can be based on a score, both for the all-comer cohort and for each triage level separately. Even in the second highest triage acuity level (ESI 2) with admission rates of 62% (our data) to 73% [35], most patients had a low risk of mortality and could theoretically have been discharged for this reason. While ESI is the best validated triage tool, risk stratification according to ESI may be of limited use because of its focus on “acuity” and the large overlap in prognosis, particularly between the highly prevalent ESI 2 and ESI 3 cohorts. Laboratory data are available for most ESI 2 and ESI 3 patients, and the TRIAL score can significantly improve the precision of risk stratification within each ESI level. In fact, the isolated laboratory model outperformed the baseline model, only to be topped by the full model and the reduced TRIAL model. While most of the parameters are known to be of prognostic value, the combination of the parameters was never assessed in an all-comer cohort. In detail, age and ESI [3,20], albumin [36], lactate dehydrogenase [37], leukocytes [11], bilirubin [11], and high levels of creatinine [38] are of prognostic value in emergency cohorts. Association between low creatinine levels and unfavorable outcome was only shown in intensive care patients [39,40]. Previous reports on the predictive value of laboratory parameters (lab) have shown similar results, but some were based only on lab [38,41], others on the combination of lab and less used triage tools [37,42], the latter showing only marginally improved predictive power if adding lab to the baseline model. Most attempts did not serve the purpose of directly supporting disposition decisions but were designed to provide relative risks, e.g., in admitted patients [12]. With our expert-based decision to choose cut-offs of 1% and 10% for admission and observation, respectively, we aimed at supporting disposition decisions directly. The advantage of this approach is the usability under different circumstances and with adjusted cut-offs. For example, in low-resource environments, cut-offs may be adjusted. In situations suitable for shared decision-making, patients may be informed about their prognosis in order to agree on disposition. The potential benefit of this approach depends less on the data set used for calculation, but in the feasibility of a risk calculation based on an all-comer cohort with lab parameters and cut-offs for disposition decisions agreed upon locally, or even individually.

Taken together, the negative predictive value of 0.99 for mortality at the 1% cut-off provides safety for early discharge. By avoiding dichotomizing variables using reference values, the loss of information can be minimized [43]. By using the TRIAL score based on points attributed to each variable, risk assessment can be made transparent and easy to understand. Finally, calibration is good up to a mortality of 20%, the cut-off for high-risk patients being set at 10%. There is also a potential to identify patients at risk at an early stage—allocating resources for their observation potentially improving outcomes.

### Limitations

This is a single-center all-comer cohort. Therefore, external validity is limited. The risk of short-term mortality is not the only reason to be admitted. In addition, pain, dyspnea, impaired function, and psychosocial aspects may mandate hospitalization. We did not investigate causes for admission systematically, e.g., by asking patients about their preferences. However, the study was designed as a proof-of-principle approach. Any institution with access to emergency, laboratory, and outcome-data could use the methods described to calculate their individual risk score in lieu of the proposed TRIAL score and decide on meaningful cut-offs for discharge, admission, and observation.

Furthermore, there is a potential selection bias due to excluding patients with no complete set of laboratory data. However, more than two-thirds (69%) of excluded patients were assigned to ESI 4 and 5. Patients without laboratory had an overall survival of 99.8%, and ESI 4/5 patients without laboratory had a survival of 100%. Doubts could also be raised about the potential use of a laboratory-based score in ESI 1 patients undergoing highly standardized life-saving interventions with subsequent observation in most cases. Similarly, ESI 4/5 patients with low admission rates may not need such a score. However, the largest groups (ESI 2 and ESI 3), in which low-risk patients dominate, may profit the most, as these patients tend to be hospitalized for fear of unfavorable outcomes. Finally, mortality could be underestimated because almost half of the patients were discharged. However, 30-day observation of the first 4190 patients showed similar results with a very low post-discharge mortality. Another limitation is the missing analysis of patients during the summer months July–September.

## 5. Conclusions

This is a proof-of-principle study aiming at disposition decision support to minimize unnecessary admissions in low-risk patients and maximize observation in high-risk patients, irrespective of diagnoses. This approach needs to be tested in an intervention trial, in order to put this evidence on a solid foundation for the benefit of patients and health care systems.

## Figures and Tables

**Figure 1 jcm-10-00939-f001:**
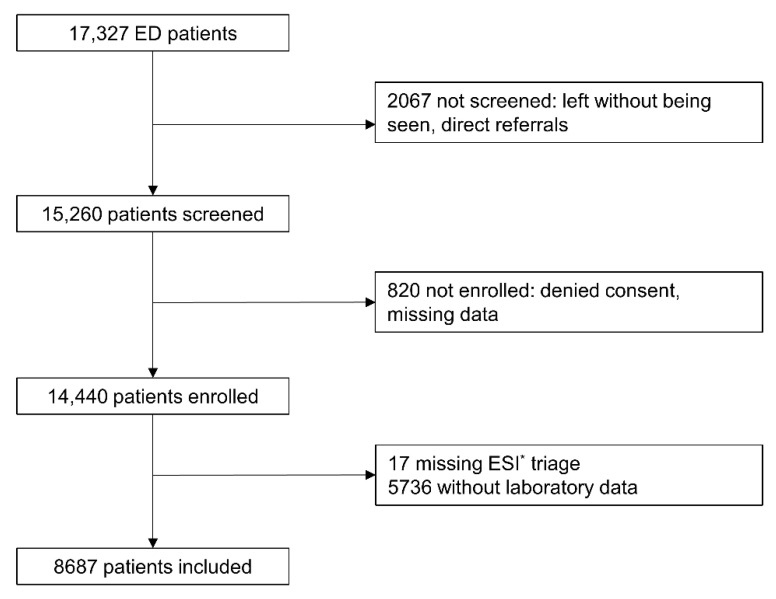
Inclusion procedure. ED: Emergency Department; * ESI = Emergency Severity Index.

**Figure 2 jcm-10-00939-f002:**
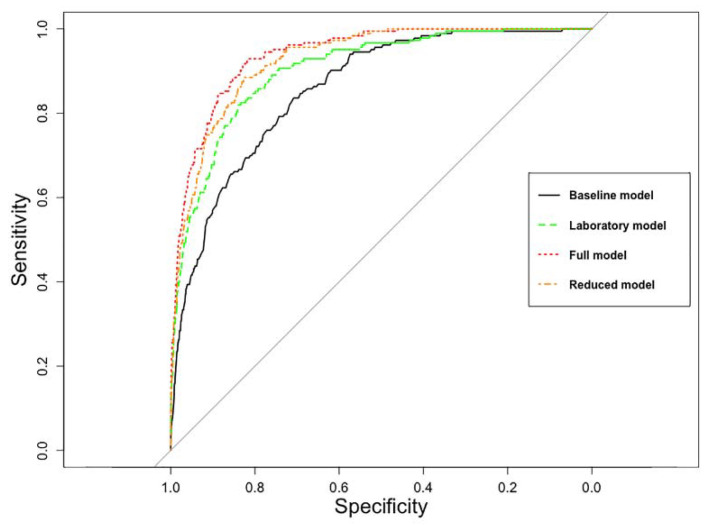
Comparison of the four multivariable logistic regression models regarding probability of in-hospital mortality. The area under the curve was 0.86 (95% CI 0.83–0.88) for the baseline model, 0.90 (95% CI 0.88–0.92) for the laboratory model, 0.94 (95% CI 0.93–0.95) for the full model, and 0.93 (95% CI 0.91–0.94) for the reduced model.

**Table 1 jcm-10-00939-t001:** Baseline characteristics and outcomes of patients included and patients without laboratory data.

	All Patients (*n* = 14,423)	Patients Included (*n* = 8687)	Patients without Laboratory Data (*n* = 5736)
Age, median (IQR *)	52 (34–72)	61 (42–78) ^§^	40 (29–56) ^§^
Male gender, *n* (%)	7547 (52.3)	4471 (51.5) ^‖^	3076 (53.6) ^‖^
ESI ^†^ level		^§^	^§^
ESI 1, *n* (%)	339 (2.4)	320 (3.7)	19 (0.3)
ESI 2, *n* (%)	3562 (24.7)	3244 (37.3)	318 (5.5)
ESI 3, *n* (%)	5939 (41.2)	4492 (51.7)	1447 (25.2)
ESI 4, *n* (%)	4243 (29.4)	623 (7.2)	3620 (63.1)
ESI 5, *n* (%)	340 (2.4)	8 (0.1)	332 (5.8)
In-hospital mortality, *n* (%)	197 (1.4)	183 (2.1) ^§^	14 (0.2) ^§^
Hospitalization, *n* (%)	4986 (34.6)	4623 (53.2) ^§^	363 (6.3) ^§^
ICU ^‡^ admsission, *n* (%)	791 (5.5)	758 (8.7) ^§^	33 (0.6) ^§^

Data are presented as medians with interquartile ranges for continuous variables and counts with percentages for categorical variables. * IQR = interquartile range, ^†^ ESI = Emergency Severity Index, ^‡^ ICU = intensive care unit, ^§^ Patients included and patients without laboratory data differ with a *p*-value <0.001, ^‖^ Patients included and patients without laboratory data differ with a *p*-value of 0.01.

**Table 2 jcm-10-00939-t002:** Demographics and outcomes of patients included stratified by ESI level (1–5).

	Age, Median (IQR *)	Male Gender *n* (%)	Hospitalization *n* (%)	ICU ^†^-Admission *n* (%)	Mortality *n* (%)
ESI 1 ^‡^	69 (55–79)	194 (60.6)	295 (92.2)	180 (56.3)	61 (19.1)
ESI 2	62 (45–78)	1758 (54.2)	1999 (61.6)	391 (12.1)	67 (2.1)
ESI 3	61 (40–79)	2192 (48.8)	2166 (48.2)	177 (3.9)	54 (1.2)
ESI 4	45 (32–66)	321 (51.5)	160 (25.7)	9 (1.4)	1 (0.2)
ESI 5	41 (28–72)	6 (75.0)	3 (37.5)	1 (12.5)	0 (0.0)

Data are presented as medians with interquartile ranges for continuous variables and counts with percentages for categorical variables. * IQR = interquartile range, ^†^ ICU = intensive care unit, ^‡^ ESI = Emergency Severity Index.

**Table 3 jcm-10-00939-t003:** TRIAL score for prediction of in-hospital mortality.

Age (years)	10	20	30	40	50	60	70	80	90	100				
Points	0	9	18	27	35	42	46	51	59	68				
ESI * (levels)	5	4	3	2	1									
Points	0	52	65	72	100									
Creatinine (μmol/L)	10	20	30	40	50	60	70	80	90	100	500			
Points	44	26	16	9	4	0	0	3	6	9	14			
LDH ^†^ (U/L)	200	300	400	500	600	700	800	900	1000	1100	1200			
Points	0	9	13	16	18	21	22	24	25	26	28			
Albumin (g/L)	50	45	40	35	30	25	20	15						
Points	0	5	10	17	22	29	37	47						
Leukocytes (×10^9^/L)	1	2	3	4	5	10	15	20	25	30	35	40	45	50
Points	23	13	7	3	0	0	4	7	10	12	13	15	16	17
Bilirubin (μmol/L)	2	5	10	20	30	40	50	60	70	80	90	100	110	120
Points	5	1	0	4	8	11	14	16	18	19	21	22	23	24
Total points	145	176												
Risk of mortality	1%	10%												

The number of points were assigned according to the relative contribution of the variables to the predicted in-hospital mortality. Test results are to be rounded to the next higher or lower value of the score. * ESI = Emergency Severity Index, ^†^ LDH = lactate dehydrogenase.

**Table 4 jcm-10-00939-t004:** Patients stratified according to ESI level (1–5) and risk of death.

	Total, *n* (%)		Non-Survivors, *n* (%)		Survivors, *n* (%)	
All	8687	(100.0)				
High risk	344	(4.0)	90	(26.2)	254	(73.8)
Intermediate risk	2478	(28.5)	86	(3.5)	2392	(96.5)
Low risk	5865	(67.5)	7	(0.1)	5858	(99.9)
ESI * 1	320	(100.0)				
High risk	173	(54.1)	53	(30.6)	120	(69.4)
Intermediate risk	127	(39.7)	8	(6.3)	119	(93.7)
Low risk	20	(6.3)	0	(0.0)	20	(100.0)
ESI 2	3244	(100.0)				
High risk	111	(3.4)	23	(20.7)	88	(79.3)
Intermediate risk	1147	(35.4)	41	(3.6)	1106	(96.4)
Low risk	1986	(61.2)	3	(0.2)	1983	(99.8)
ESI 3	4492	(100.0)				
High risk	60	(1.3)	14	(23.3)	46	(76.7)
Intermediate risk	1184	(26.4)	37	(3.1)	1147	(96.9)
Low risk	3248	(72.3)	3	(0.1)	3245	(99.9)
ESI 4	623	(100.0)				
High risk	0	(0.0)	0	(0.0)	0	(0.0)
Intermediate risk	20	(3.2)	0	(0.0)	20	(100.0)
Low risk	603	(96.8)	1	(0.2)	602	(99.8)
ESI 5	8	(100.0)				
High risk	0	(0.0)	0	(0.0)	0	(0.0)
Intermediate risk	0	(0.0)	0	(0.0)	0	(0.0)
Low risk	8	(100.0)	0	(0.0)	8	(100.0)

Patients are stratified according to ESI level (1–5) and their predicted risk of in-hospital mortality (low, intermediate, and high risk). Furthermore, the number and percentage of patients who died in-hospital in each risk group is shown per ESI level. * ESI = Emergency Severity Index.

## Data Availability

The data presented in this study are available on request from the corresponding author.

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
