# Peer review of "Disposition Decision Support by Laboratory Based Outcome Prediction"

_jcm, 2021, doi:10.3390/jcm10050939_

Round 1

Reviewer 1 Report

This is an interesting effort by the authors to find a method of augmenting the predictability of need for hospitalization by using easily obtainable laboratory parameters in addition to the usual emergency department triage level assignment.  I agree that such efforts are worthwhile given the hospital capacity issues that are present in many facilities, and the desire to avoid expensive hospitalization if it is not medically necessary.

The author's approach is solid and the analysis appropriate.  They appropriately exclude patients without lab tests being obtained in the normal course of care, and address that decision adequately in the discussion section.

As someone who works in clinical informatics and sees lots of papers on predictive algorithms, my natural inclination is to jump right to the need to see something like this implemented in a clinical workflow to determine if there is a material impact on actual clinical decision making.  However, the authors are very clear that this is a descriptive effort of the development of such a predictive algorithm, and that "real-world" implementation is still needed to assess what value it might have in real life.

The key issue with an algorithm like this -- and the authors do address this limitation in the discussion -- is that the decision to hospitalize patients is incredibly subjective, and I am somewhat skeptical that a tool like this is going carry enough weight to cause a medical practitioner in the emergency department to change their mind regarding hospitalizing a patient for any of the subjective reasons the authors list.

All that, however, is more the next phase of evaluation of this tool.  Here, the authors have done a fine job of outlining the algorithm, justifying the inclusion and exclusion of the various parameters, emphasizing the need for an implementation study as a next step, and outlining the limitations of what a tool like this can be expected to do.

I commend the authors in targeting their work to develop a tool to support an operational workflow (disposition from the emergency department), as such tools are needed in clinical work today. 

I have no major recommendations for revision or editing to this manuscript.

Reviewer 2 Report

Thank you for allowing me to review this beautiful manuscript! As a first feeling, I was impressed to read that you obtained an area under ROC curve larger than 0.9 in an observational study. I then realized that the authors use a well-defined and validate tool (ESI scale) and their goal is to improve it by including additional predictors (laboratory values and age). They actually start with an AUC of 0.86 with model A, but the addition of laboratory values still allows to improve the prediction.

In general, the methodology is well described, but there are still some unclear aspects. In the presentation of the models, I do not really understand how model B is built. The authors mention cubic splines, in order to account for nonlinear relationships to in-hospital mortality. The problem is that as a reader, I have the feeling that in-hospital mortality is a binary (yes/no) variable, so I do not understand this choice of cubic spline. Is that related to figure A1? In the beginning of section 2.5.2, they mention nonlinear logistic regression and again, I do not understand why the models are nonlinear. Even with splines, we are still in a linear framework.

The model D is also unclear to me. Backward elimination is a nice procedure to select important variables and prevent from collinearity. But some precisions are missing. On which criterion is done the selection (AIC, BIC, F-test, likelihood ratio test,...)? And which cut-off?

The authors performed bootstrap to prevent from overfitting problems. To my opinion, it is not enough. They would have better to split randomly the sample in two parts, a training set to fit the model and then a test set to validate it. Overfitting can be detected for example by a large Mean Squared Error on the test set. The ideal solution would be to consider data from another hospital and try the model on this new hospital. This would significantly improve the quality of the model, and of the paper.

Considering "only" one hospital is a limitation of the study. Another clear limitation is the high amount of data that are not considered, due to missing values (5736 patients among 14440, this is a very large proportion). This problem is maybe not a problem, since the authors mention that "patients without laboratory data were younger, had lower acuity ESI levels and better outcomes". To ensure this, the authors may try to impute missing data, and see how robust are the models.

In section 2.1, the authors mention the wish to consider "potential seasonal effects", but February is overrepresented and Summer is missing. Moreover, this aspect is not discussed in the paper. Did they try to include it in the analyses? Are there some differences e.g. between Autumn and Spring?

Round 2

Reviewer 2 Report

I want to thank the authors for having taken into account the remarks I wrote in my first review. To my opinion, the paper quality clearly improved. The methodology is better described. Moreover, the 10-fold cross-validation analysis is a very good idea to ensure the quality of the models. I congratulate the authors for their work and for the promptness with which they developed this second version.